# In-Treatment Changes in Quality of Life-Related Variables in Therapeutic Communities for Cocaine Abusers: Are These Changes Associated with Clinical Outcomes?

**DOI:** 10.3390/ijerph18147442

**Published:** 2021-07-12

**Authors:** Francisco González-Saiz, Esperanza Vergara-Moragues

**Affiliations:** 1Departamento de Neurociencias, Área de Psiquiatría, Universidad de Cádiz, Hospital Universitario de Jerez, 11407 Cádiz, Spain; 2Instituto de Investigación e Innovación Biomédica de Cádiz (INiBICA), 11009 Cádiz, Spain; espeverg@ucm.es; 3Centro de Investigación Biomédica en Red de Salud Mental (CIBERSAM), 28029 Madrid, Spain; 4Departamento de Psicobiología y Metodología en Ciencias del Comportamiento, Universidad Complutense de Madrid (UCM), 28040 Madrid, Spain

**Keywords:** therapeutic communities, quality of life, addiction, treatment outcome, reliable change index

## Abstract

Background: Few studies have explored changes in quality of life during the first three months of admission to a therapeutic community for addictions. The objective of this study was to determine the relationship between these changes and treatment outcomes at discharge. Methods: We undertook a prospective longitudinal study of 142 cocaine-dependent patients treated at a therapeutics community. All of these patients reached the 3-month evaluation and were subsequently followed until discharge. All participants completed the following measures: Health-Related Quality of Life for Drug Abusers Test; Beck Depression Inventory; State–Trait Anxiety Inventory; Opiate Treatment Index; Dual Diagnosis and Discharge Checklist. Results: At the 3-month assessment, scores on the Health-Related Quality of Life for Drug Abusers Test had increased significantly (Cohen’s d: 0.92), while scores on the Opiate Treatment Index (Cohen’s d: 0.86) and Beck Depression Inventory (Cohen’s d: 0.20) scales both decreased significantly. A higher proportion of the patients considered to have achieved “clinically relevant” treatment outcomes at discharge versus those without clinically relevant outcomes were considered “recovered” according to the Reliable Change Index. Conclusions: An improvement in quality of life-related variables from baseline to the 3-month assessment was associated with better outcomes at discharge from the therapeutic community. The findings of this study may help us to optimise therapeutic interventions.

## 1. Introduction

Cocaine use is a major public health problem [1]. Worldwide, there are an estimated 19 million cocaine users, corresponding to 0.4% of the world’s population between the ages of 15 and 64. In Europe, approximately 1.4% of adults are cocaine users [1]. Healthcare-related indicators and mortality data suggest that the use of cocaine has increased in recent years [2,3]. Cocaine use is associated with a high prevalence of social, family, legal, physical and mental health problems [1,4,5,6,7,8,9].

At present, there is no consensus regarding the most effective treatment strategy (pharmacological and psychosocial) for cocaine use. Although many patients are treated at outpatient centres, relapse rates are high [10,11,12,13,14,15,16]. A growing body of research suggests that admission to a therapeutic community (TC) may be an effective intervention for these patients [17]. In fact, the scientific literature has consistently demonstrated the efficacy and effectiveness of TC programmes for the treatment of substance abuse in terms of decreased cocaine use and criminal activity, as well as better mental health and social functioning [17,18,19,20,21]. However, relatively few studies have evaluated the influence of changes in quality of life (QoL) measures on treatment outcomes in patients treated at TCs [22,23,24,25,26]. This lack of data is notable, especially given that—from a recovery-oriented perspective—a better QoL should be considered a key aim of treatment in a TC [20,26].

The effectiveness of the treatment programme depends on the patient’s individual characteristics at admission and on the treatment process itself [17,20,27,28]. In most studies, the best predictor of the effectiveness of TCs is the length of stay (treatment retention) [29,30,31,32,33,34,35,36,37,38], mainly because a longer stay implies that the patients will receive more therapeutic interventions, which may increase the likelihood of positive change [28,39,40]. Numerous studies have found that the minimum length of stay needed to obtain clinically relevant results is approximately three months [34,37,41,42,43,44,45,46].

Despite the widely recognised importance of reaching this minimum length of stay, relatively few studies have explored the clinical changes in psychological functioning or QoL status over this 3-month period. Nonetheless, several studies have observed a reduction in depressive symptoms and general or social anxiety [22,23,47,48,49,50] and a better QoL [22,23,25]. However, the aforementioned studies measured the improvement in QoL-related variables as differences in group means from baseline to the 3-month assessment, rather than assessing the changes experienced by each patient individually. From a patient-centred clinical perspective, it is becoming increasingly important to measure psychosocial treatment outcomes at the individual level [51]. Moreover, to the best of our knowledge, the association between changes in psychological functioning and QoL during the first three months of TC and clinical outcomes at discharge has not been evaluated to date. 

In this context, the main objective of this study was to determine the relationship between changes in four QoL-related variables (addiction-related QoL, state anxiety, depression, and general health status) after three months of treatment and treatment outcomes at discharge in patients with cocaine use disorder in a therapeutic community. We hypothesised that patients who remained in the TC for ≥ three months would experience a significant improvement in QoL-related variables and that this improvement would be associated with treatment outcomes at discharge.

## 2. Materials and Methods

### 2.1. Design and Setting

This was a prospective, longitudinal study involving cocaine-dependent patients treated at one of six public TCs for the treatment of substance abuse in Andalusia, Spain (Cartaya, Almonte, Mijas, Los Palacios, La Línea and Tarifa). All of these centres offer the same treatment programme based on a multidisciplinary approach, consisting of cognitive behavioural therapy, psychoeducation, occupational therapy, and pharmacological treatment [52]. The duration of the treatment programme is individualised, but usually lasts from 6 to 9 months. These six TCs form part of a public network of psychosocial care for drug addiction, which includes outpatient care centres, homes with different supervision levels, and employment support programmes. All patients admitted to the TC are referred by outpatient care centres. Upon discharge from the TC, follow-up care continues at the original outpatient centre. The most common criteria for admission to the TC are the inability to maintain abstinence in the outpatient setting and the perception—by the patient and/or clinician—that more intensive and continuous treatment is needed. Admission to the TC is always voluntary. 

### 2.2. Participants

A consecutive sampling technique was used to recruit participants for this study, including all patients who met the selection criteria. The inclusion criteria were: (a) fulfilment of DSM-IV diagnostic criteria for cocaine dependence [53], (b) age ≥ 18 years, (c) ability to read and write, and (d) signed informed consent. The exclusion criteria were: (a) the presence of an organic cognitive impairment that impedes clinical evaluation and (b) imprisonment pending and expected within six months of admission. Considering the study objectives, the final sample included the 142 patients who reached the 3-month post-admission evaluation, out of a total of 226 subjects who met the study selection criteria. The sample size calculation was based on data provided by Nunes et al. [54], assuming a power (1-α) of 0.9 and a significance level of *p* ≤ 0.05. The distribution of participants by treatment centre was as follows: Tarifa (22%), Almonte (21.6%), Los Palacios (21.5%), Cartaya (17%), La Línea (13.8%), and Mijas (4.1%). All eligible patients were invited to participate and those who met the selection criteria, agreed to collaborate, and signed an informed consent form were included. A total of 84 patients were excluded from the analysis due to a failure to reach the minimum follow-up of 3-months; of these 84 patients, 61.9% (52), voluntarily withdrew from the TC and the other 38.1% were discharged early (32) for violating the rules of the TC. The sociodemographic and clinical characteristics of the final sample (*n* = 142) are shown in Table 1, column 3. Most of the participants were males (92.3%). The mean age was 34.6 years. Most participants were single (64.1%) with a secondary educational level (58.5%). The majority of the participants were unemployed (68.3%) and 56.3% had a criminal record. Most patients (80.3%) had received prior treatment for substance use disorder and 40.8% had been previously treated for a mental disorder. In terms of the substance use disorder, most patients (66.4%) were speedball users (smoked/inhaled), while 33.6% were cocaine users (22.4% used powder cocaine and 11.2% freebased) without heroin use. The mean duration of cocaine use was 10 years, with a relatively severe dependence at admission. Other substance use disorders, in addition to cocaine dependence, included addiction to heroin (67.2%), cannabis (58.9%), and alcohol (58.3%). Two-thirds (66.2%) of the patients had at least one co-occurring psychiatric disorder (dual diagnosis). The mean baseline scores on the QoL, state anxiety, depression and general health status scales were consistent with the average scores typically observed in this patient population 

### 2.3. Measurement Instruments

***Health-Related Quality of Life for Drug Abusers Test*** (HRQOLDA) [55]. This is a specific QoL assessment instrument for drug-dependent patients based on the bi-axial concept of addiction. The HRQOLDA evaluates the physical and psychosocial effects that the substance of abuse produces on health and daily functioning, as well as individual responses to these effects. It is a 22-item, self-administered symptom scale with responses coded on a 5-point Likert scale (1 to 5). Higher total scores indicate better health-related QoL. We used the original Spanish version of this instrument. In our study, the internal consistency was acceptable (Cronbach’s α = 0.81).

***Beck Depression Inventory*** (BDI-I) [56,57]. This is a self-administered symptom scale designed to assess depression severity. We used the 13-item version. Responses are given on a 4-point Likert-type scale (0 to 3). The total maximum total score is 39, with higher total scores indicating greater severity. Although this tool includes cut-off points to stratify patients by severity level, we used the dimensional evaluation model to compare changes in the total scores from baseline to the follow-up assessments. We used the Spanish version of the BDI-I [58]. In our study, Cronbach’s α was 0.91.

***State–Trait Anxiety Inventory*** (STAI) [59]. This is a self-administered inventory that assesses anxiety-related symptoms. It consists of two scales that measure different but related aspects: state and trait anxiety. We used the anxiety-state scale, which assesses anxiety as the immediate emotional attitude to a specific situation, which is the component of anxiety with the greatest variability. The STAI contains 20 items with responses given on a four-point Likert scale (0 to 3). Higher scores indicate greater severity. We used the Spanish version of this scale [60]. In this study, Cronbach’s α was 0.89.

***Opiate Treatment Index (OTI)*** [61]. The OTI is a semi-structured clinical interview designed to evaluate problems related to substance use and treatment outcomes. It includes six subscales, each of which provides a measure of severity or impairment in the corresponding domains, as follows: drug use; HIV risk-taking behaviour; social functioning; criminality; health; and psychological functioning. On all subscales, higher scores indicate greater dysfunction or severity. We used the health symptom subscale (OTI-HSS) of the Spanish version of this instrument [62,63] to evaluate physical health status as an indicator of QoL. The health subscale provides a checklist subdivided into eight sections, each of which measures a set of health-related symptoms grouped by a major organ system, as follows: general, injection-related problems, cardiorespiratory, genito-urinary, gynaecological, musculoskeletal, neurological, gastro-intestinal. In our study, Cronbach’s alpha was 0.72.

***Psychiatric Research Interview for Substances and Mental Diseases*** (PRISM) [64]. The PRISM is a semi-structured clinical interview, specifically designed for the diagnosis of psychiatric comorbidity in patients with substance use disorders. PRISM enables clinicians to differentiate between the expected effects of intoxication/withdrawal, independent (primary) disorders, and substance-induced psychiatric disorders. The PRISM measures the main Axis I (mood, anxiety, psychotic, and eating disorders) and Axis II (borderline personality and antisocial personality disorder) diagnoses of the DSM-IV. The diagnoses distinguish between two time frames; (a) “current disorder”, defined as a disorder present in the previous 12 months, and (b) “lifetime disorder”, defined as a disorder diagnosed at a previous point in time (which, therefore, also includes “current disorders”). We used the first Spanish version of PRISM, which has been shown to present adequate psychometric properties [65]. Since the first version of PRISM does not evaluate attention-deficit hyperactivity disorder (ADHD), we used another instrument to check for the presence of ADHD (Conners’ Adult ADHD Diagnosis Interview for DSM-IV—CAADID) [66]. Finally, we developed a dichotomous variable, “dual diagnosis”, understood as the presence of at least one diagnosis in the PRISM (lifetime diagnoses) and CAADID interviews. We also used other parts of PRISM to measure sociodemographic variables and variables related to substance use. 

***Conners’ Adult ADHD Diagnostic Interview for DSM-IV*** (CAADID) [66]. The CAADID is a semi-structured clinical interview used to diagnose ADHD according to DSM-IV criteria. The first section assesses symptoms and the second section explores all the symptoms of this disorder, both in adult life and during childhood and adolescence. We used the Spanish version of this instrument [67], which has adequate psychometric properties [68]. 

***Discharge Checklist*** The clinical staff at the six public TCs involved in this study systematically evaluate all patients prior to discharge using a form known as the Discharge Checklist. The purpose of this tool is to assess the care process and treatment outcomes, understood as the degree to which the treatment objectives established for each patient at admission were met. The Discharge Checklist consists of 11 blocks, grouped into three main domains (psychological, socio-educational, and health). Based on these data and clinical impressions, treatment outcomes are globally classified into one of three categories: (1) “No changes”: the patient’s stay in the TC was short and response to treatment was either not assessable or the patient’s attitude was persistently challenging, aggressive or non-collaborative; (2) “Minimal changes”: minimal but insufficient clinical progress on at least one of the three main domains; and (3) “Clinically relevant changes”: there was sustained and progressive improvement in one or more of these three domains and the treatment objectives were met. All of the TCs share a common therapeutic programme and standardised clinical guidelines, and all use the same evaluation procedures. The clinical staff was unaware that this information would later be used by the research team as a treatment outcome measure. For the purposes of this study, we reduced these three outcome categories to just two categories, as follows: (1) “Non-clinically relevant outcomes”, which comprised patients classified as showing “no changes” or “minimal changes”, and (2) “Clinically relevant outcomes”, which included all patients who showed “clinically relevant changes”.

### 2.4. Procedures

To identify study candidates, the clinical team at the participating TCs assessed the patients admitted each week. Potentially eligible patients were informed of the study objectives and asked to participate. Full anonymity and confidentiality were assured. All patients were evaluated by an external psychologist (E.V.M.) with an undergraduate degree in psychology and a master’s degree in clinical psychology. The external evaluator received specific training in the administration of the clinical interviews for all the semi-structured instruments used in the study. The baseline evaluation was carried out approximately 15 days after admission to minimise the impact of intoxication and withdrawal symptoms, as these could have influenced the clinical evaluation. The interviews were conducted in an office at the participating TCs. Participants did not receive any financial compensation or any other type of privilege for participating in the study. All candidates were informed that non-participation in the study would not affect the quality of the treatment received at the TC and that they could refuse consent and withdraw from the study at any time without any negative consequences. The length of stay at the TC, defined as the time from admission to discharge, was registered for all patients. All clinical data were anonymised for the analysis to ensure data confidentiality. All of the ethical standards of the Declaration of Helsinki for research with people were followed. The research protocol was approved by the Research Ethics Committee at the University of Granada.

### 2.5. Statistical Analysis

For the descriptive analyses, we calculated means with standard deviations (SDs) or percentages, as appropriate. To evaluate the hypothesis, we calculated the differences in means using Student’s *t*-test for quantitative variables and χ^2^ for qualitative variables. The cut-off for statistical significance was *p* ≤ 0.05. We applied a mean difference analysis (*t*-test for paired samples) to assess changes in group means between the baseline and the 3-month assessments. To evaluate “clinically significant changes” in the QoL-related variables from baseline to the 3-month assessment individually for each patient, we used the Reliable Change Index (RCI) developed by Jacobson and Truax [51]. The RCI defines a “clinically significant change” as a change over time from impaired to normal functioning. Statistically, this requires the establishment of a cut-off point on the RCI to distinguish between “normal” and “dysfunctional” populations. To be considered a clinically relevant change, the RCI score must be ≥two SDs above the mean value. All statistical analyses were performed with the SPSS statistical package, v.22 (IBM Corp., Armonk, NY, USA).

## 3. Results

### 3.1. Baseline Comparison: Patients Who Reached the 3-Month Assessment versus Early Dropouts

Of the 226 patients admitted to the study (Table 1, columns 2 to 5), 142 reached the 3-month assessment (“completers”), translating into a cumulative retention rate of 62.8%. The remaining 37.2% abandoned treatment early (“non-completers”). The “completer” group was comprised of a significantly higher proportion of single patients and a lower percentage of retired/disabled patients compared to the non-completers. In addition, the completer group had lower mean scores on the STAI state anxiety and BDI depressive symptom scales. No between-group differences were observed in the scores of the HRQOLDA and OTI-HSS scales, nor in the prevalence of dual diagnosis. No differences between the groups were observed with regard to substance use-related variables.

### 3.2. Comparative Analysis: QoL-Related Variables from Baseline to the 3-Month Assessment

As shown in Table 2, at the 3-month assessment, the group means on the OTI-HSS scale (general health status) and BDI (depressive symptoms) both decreased significantly while HRQOLDA QoL scores increased significantly. The mean difference in the STAI state anxiety scale scores did not change significantly from baseline. The effect size values of these changes (Cohen’s d) were relevant for the changes observed in both the OTI-HSS and HRQOLDA scales.

### 3.3. Comparison Analysis: Scores on the QoL-Related Instruments at the 3-Month Assessment and Clinical Impression of Treatment Outcomes at Discharge

Table 3 shows the comparison between QoL-related variables at the 3-month assessment and the treatment outcomes at discharge. The subgroup of patients presenting “clinically relevant outcomes” at discharge had significantly lower scores on both the BDI and STAI scales and significantly higher scores on the HRQOLDA scale at the 3-month assessment compared to those with “non-clinically relevant outcomes”. No statistically significant between-group differences were observed on the OTI-HSS scores.

### 3.4. Comparison Analysis: Reliable Changes in QoL-Related Variables from Baseline to the 3-Month Assessment and Clinical Impression of Treatment Outcomes at Discharge

Table 4 (first column) shows the result of the descriptive analyses of the individual (grouped) changes experienced by patients during the first three months of treatment for four QoL-related variables. The individual changes in each variable were grouped into five categories based on the RCI values. Nearly 80% of patients presented some improvement in OTI-HSS scores, with 51.4% considered “recovered”. On the BDI scale, 56.3% showed an improvement and 37.4% were considered “recovered”. On the STAI state anxiety scale, 48.6% of patients presented some improvement, with 28.9% “recovered”. On the HRQOLDA scale, 83.1% of patients improved, with 61.3% “recovered”. The magnitude of effect sizes (Cohen’s d coefficients) paralleled these results (Table 2).Table 4 (columns 2 and 3) shows the results of a descriptive analysis comparing the change categories for each QoL-related variable between patients with “clinically relevant” treatment outcomes versus those with non-clinically relevant outcomes at discharge. As the table shows, the proportion of patients with clinically relevant outcomes at discharge who were classified as “recovered” on the BDI, STAI and HRQOLDA scales at the 3-month assessment was higher than in the non-clinically relevant outcome group. No differences on the OTI-HSS scale were observed between these groups. These findings were largely consistent with the results of the group data analyses of these scales (Table 3).

### 3.5. Association between Reliable Changes in QoL-Related Variables at the 3-Month Assessment and Mean Values at Baseline

Figure 1 shows, for each scale, reliable change outcomes versus mean baseline scores. As shown in Figure 1A, the patients who experienced the greatest improvement on the OTI-HSS general health scale (“Improved”, “Recovered1SD”, “Recovered2SD”) had higher baseline scores on this scale (i.e., worse health status at baseline). By contrast, the patients with lower baseline scores (i.e., better health) presented less improvement or even a small decrease. Similar results were observed for all other QoL-related variables (Table 1B–D).

## 4. Discussion

The findings of this study support our initial hypothesis that the quality of life of cocaine-dependent patients would improve after three months of treatment in a therapeutic community, and that this initial improvement would be associated with better treatment outcomes at discharge. In this work, following the recommendations of Diener and Ryan [69], we used a broad concept of QoL that included variables related to psychological well-being, such as anxiety and depression, as well as general health status in association with substance use. We found that health-related QoL—evaluated with an instrument specifically designed to assess this dimension in drug addicts—improved significantly during the 3-month treatment period with a large effect size, a finding that is consistent with a previous study carried out by our group in opiate-dependent patients treated with buprenorphine, who showed improvement even before the 3-month assessment [22,23]. Our findings provide further support for the proposal made by Broekaert et al. [26], who stated that improvements in QoL should be considered a fundamental aim of TC programmes for addictions and an indicator of treatment response. 

We observed a significant reduction in depressive symptom severity, although the magnitude of improvement was small. Similar findings were reported in three other studies that assessed 3-month outcomes after admission to a TC. Two of those studies—those by Egelko et al. [48] and McKee et al. [50]—used the BDI scale to assess depressive symptom severity, while the other [49] administered the depression subscale of the Minnesota Multiphasic Personality Inventory-2. We also observed a significant reduction, with a relevant effect size, in the severity of drug use-related health problems, consistent with our previous findings in opiate-dependent patients three months after admission to a therapeutic community [22]. 

We did not observe any significant reductions in anxiety symptoms during the first three months of treatment, a finding that stands in contrast to the positive results reported by several authors [47,48,49,50]. This discrepancy between our findings and those other studies could be, at least partially, due to differences in the instruments used to measure anxiety (ours was the only study to use the STAI) and/or to clinical differences in the study samples.

The results of this study demonstrate that the clinical improvements observed after three months of treatment in a TC are associated with better treatment outcomes at discharge. To the best of our knowledge, this is the first study to investigate this association in this setting, and the first to assess reliable changes at the individual level. None of the four studies cited above evaluated the association between psychological changes during the first three months of treatment and the final outcomes at discharge from the TC. By contrast, we evaluated this association from two different analytical perspectives. First, we assessed changes in these variables based on an analysis of group means, finding that patients who presented clinically relevant outcomes at discharge presented a better QoL and fewer anxiety and depressive symptoms at the 3-month assessment (Table 3). However, it should be noted that using statistical significance as the only measure of the impact of psychosocial interventions has important limitations. First, since this measure is based on group means, it does not provide information on individual variability. Second, it does not address the question of clinical significance [70]. To overcome this limitation, Jacobson and Truax [51] developed the Reliable Change Index to measure clinical significance. The RCI provides valuable information and can be applied to complement the findings of group mean comparisons. In the present study, we applied the RCI to provide a second analytical perspective, obtaining results that were consistent with the group means analysis, but with the added advantage of being more clinically applicable. When we compared patients with “clinically relevant” versus “non-clinically relevant” outcomes at discharge, the first group had a higher proportion of patients that were considered “recovered” (defined as an improvement of two standard deviations versus the mean) in the QoL, anxiety and depression scores at the 3-month assessment. Although the criteria used to define “recovery” for a given variable (two SDs from the mean) is the standard cut-off point, this cut-off point may be too demanding—as Iraurgi-Castillo [70] observed—when evaluating clinical significance in severely ill patients, such as drug dependent patients and/or those with mental disorders. Therefore, although a cut-off point of two standard deviations better discriminates between “recovered” patients with better treatment outcomes at discharge than a one-SD cut-off point (Table 4), the latter point is less demanding and more consistent with the expected change in seriously ill patients, such as those with addictive disorders.

We did not find any association between the severity of drug-related health problems and clinical outcomes at discharge from either of these two analytical perspectives. Interestingly, the magnitude of recovery and improvement was linearly related to the baseline levels on each scale (Figure 1). Logically, patients with the highest levels of depressive symptoms at baseline were precisely those that experienced the most clinically significant changes, while patients with low BDI scores (i.e., “absence of depression”) did not experience a large reduction in symptoms, which, in some cases, worsened slightly during their 3-month stay (Figure 1A). Based on our clinical experience routinely assessing treatment outcomes at discharge from the TC, it is common sense not to expect major improvements in depressive symptoms in patients with low baseline symptoms.

This study has several limitations and thus the conclusions should be interpreted and extrapolated to other healthcare settings cautiously. First, the study was conducted in a group of TCs that form part of a network of coordinated public services in which patients continue treatment after discharge at outpatient treatment centres. In addition, we applied the “in-treatment follow-up outcome measures” proposed by McLellan [71], in which the clinical evaluation at discharge is considered a proxy outcome measure. Therefore, our results cannot be extrapolated to other healthcare models. Second, this was not a randomised clinical trial and there was no control group. Therefore, we cannot conclude that the changes observed in QoL-related variables during the first 3 months of treatment are strictly due to the therapeutic interventions, nor can we control for a possible regression to the mean for the changes observed on these scales. Finally, our analysis included only the 142 patients who reached the follow-up evaluation. In the initial comparison, compared to the group of patients who dropped out of TC, these patients had more symptoms of anxiety and depression, as well as two other sociodemographic risk factors that could potentially influence the results. Future studies should take these limitations into consideration. By contrast, this study has two main strengths. First, treatment outcomes were determined by an indicator used in the routine clinical evaluation by the clinical staff based on the “in-treatment follow-up outcome measures” model [71]. Second, we evaluated the clinical significance of individual patient changes on these QoL-related variables using the RCI developed by Jacobson and Truax [51] as a complement to the group mean comparisons. Third, it is important to keep in mind that we used self-administered scales (e.g., BDI and STAI) in this study to evaluate depression and anxiety. Therefore, we cannot rule out a subjectivity bias. To reduce the risk of this type of bias, future studies should include objective assessment instruments, such as the Hamilton Depression Rating Scale (HDRS) and/or the Hamilton Anxiety Rating Scale (HDRS), which are clinician-administered scales.

Future avenues of research should seek to elucidate the mechanisms underlying the changes experienced by patients during admission to a TC. At present, although clear changes were observed, the underlying mechanisms remain uncertain. In this regard, it would be valuable to measure these changes and, in particular, explore the association between these changes and certain types and intensities of therapeutic interventions, similar to the methods used by other authors in other addiction programmes [40,72]. Finally, we believe—as our data seem to support—that QoL should be a key outcome variable in the treatment of patients admitted to a TC for substance use disorder.

## 5. Conclusions

The findings of this study show that most cocaine-dependent patients admitted to a therapeutic community present significant improvements in health-related quality of life, depressive symptoms, and health-related problems after three months of therapy. Patients with clinically relevant outcomes at discharge had higher levels of recovery in terms of QoL and anxiety and depressive symptoms at the 3-month assessment after admission. The findings of this study help us to better understand the changes experienced by patients in the first few months after admission to a TC and could help to optimise therapeutic interventions.

## Figures and Tables

**Figure 1 ijerph-18-07442-f001:**
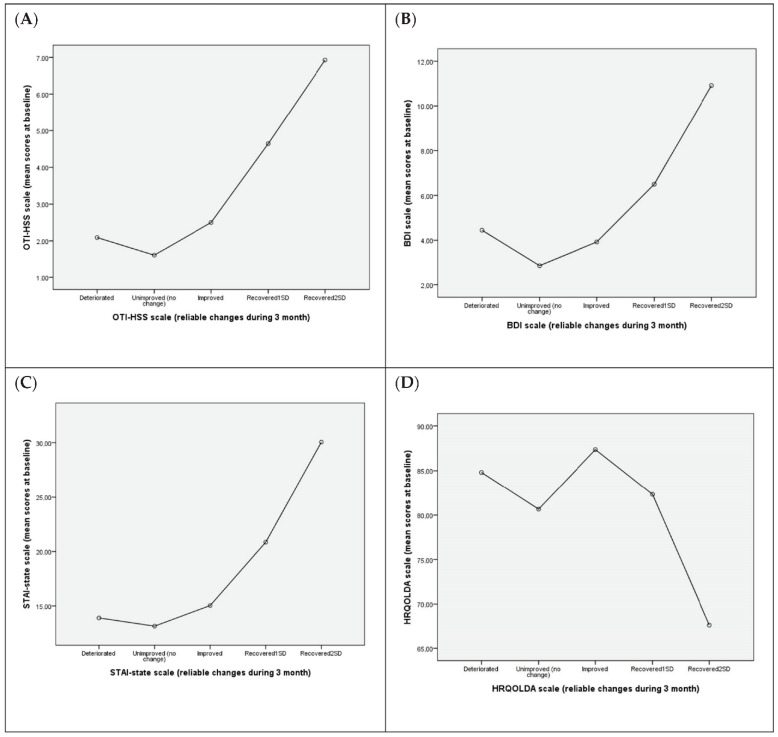
Association between reliable changes in QoL-related variables at the 3-month assessment and mean values at baseline. (**A**) OTI-HSS: Opiate Treatment Index-Health Symptom Scale; Worsened. (**B**) Beck Depression Inventory; (**C**) STAI-state = State–Trait Anxiety Inventory; (**D**) HRQOLDA = The Health-Related Quality of Life for Drug Abusers Test.

**Table 1 ijerph-18-07442-t001:** Comparison between patients who reached the 3-month follow-up evaluation and early dropouts.

	Total Sample (*n* = 226)	Early Dropout Group (*n* = 84)	3-Month Follow-Up Group (*n =* 142)	Statistical	*p*-Value
**Age**, M(SD)	34.66 (7.55)	34.73 (7.93)	34.62 (7.35)	t(224) = 0.102	0.919
**Gender**, *n* (%)					
Women	20 (8.8)	9 (10.7)	11 (7.7)	χ^2^(1) = 0.576	0.448
Men	206 (91.2)	75 (89.3)	131 (92.3)		
**Marital status**, *n* (%)					
Single	129 (57.1)	38 (45.2)	91 (64.1)	χ^2^(2) = 8.869	0.012
Married/Partner	43 (19.0)	23 (27.4)	20 (14.1)		
Divorced/Separated/Widowed	54 (23.9)	23 (27.4)	31 (21.8)		
**Educational level**, *n* (%)					
Primary school	99 (43.8)	40 (47.6)	59 (41.5)	χ^2^(1) = 0.790	0.374
Secondary school	127 (56.2)	44 (52.4)	83 (58.5)		
**Work status**, *n* (%)					
Employed	46 (20.4)	13 (15.5)	33 (23.2)		
Unemployed	152 (67.3)	55 (65.5)	97 (68.3)	χ^2^(2) = 6.410	0.041
Retired/Disabled	28 (12.4)	16 (19.0)	12 (8.5)		
**Criminal history**, *n* (%)	132 (58.4)	52 (61.9)	80 (56.3)	χ^2^(1) = 0.673	0.412
**Previous treatments for SUD ^a^**, *n* (%)	181 (80.1)	67 (79.8)	114 (80.3)	χ^2^(1) =0.009	0.925
**Previous treatments for MD ^b^**, *n* (%)	98 (43.4)	40 (47.6)	58 (40.8)	χ^2^(1) = 0.986	0.321
**Age at start of cocaine dependency**, M(SD)	24.27 (7.02)	23.86 (7.92)	24.52 (6.45)	t(224) = −0.682	0.496
**Severity of cocaine dependency**, M(SD)	6.50 (0.87)	6.36 (1.02)	6.57 (0.75)	t(224) = −1.747	0.108
**Lifetime SUD ^a^**, *n* (%)					
Heroin	152 (67.3)	59 (70.2)	93 (65.5)	χ^2^(1) = 0.540	0.463
Alcohol	133 (58.8)	51 (60.7)	82 (57.7)	χ^2^(1) = 0.192	0.661
Cannabis	134 (59.3)	53 (63.1)	81 (57.0)	χ^2^(1) = 0.801	0.371
Sedatives	73 (32.3)	30 (35.7)	43 (30.3)	χ^2^(1) = 0.712	0.399
Hallucinogens	31 (13.7)	12 (14.3)	19 (13.4)	χ^2^(1) = 0.037	0.848
Stimulants	32 (14.2)	13 (15.5)	19 (13.4)	χ^2^(1) = 0.191	0.662
**OTI-HSS ^c^**, M(SD)	3.93 (3.85)	3.98 (4.92)	3.90 (3.07)	t(224) = 0.134	0.881
**BDI ^d^**, M(SD)	6.67 (5.64)	7.75 (6.64)	6.03 (4.88)	t(224) = 2.225	0.041
**STAI-state ^e^**, M(SD)	18.66 (11.18)	20.94 (12.58)	17.32 (10.06)	t(224) = 2.374	0.026
**HRQOLDA ^f^**, M(SD)	78.58 (14.16)	78.73 (13.86)	78.49 (14.39)	t(224) = 0.127	0.900
**Dual diagnosis ^g^**, *n* (%)	156 (69.0)	62 (73.8)	94 (66.2)	χ^2^(1) = 1.431	0.232

Note: SUDs were diagnosed with DSM-IV-R. ^a^ SUD: Substance Use Disorders; ^b^ MD: Mental Disorders; ^c^ OTI-HSS: Opiate Treatment Index-Health Symptom Scale at baseline; ^d^ BDI = Beck Depression Inventory; ^e^ STAI-state = State–Trait Anxiety Inventory; ^f^ HRQOLDA = The Health-Related Quality of Life for Drug Abusers Test; ^g^ Dual diagnosis = at least one mental disorder diagnosed by PRISM and CAADID.

**Table 2 ijerph-18-07442-t002:** Comparison of changes in quality of life-related variables between baseline and the 3-month follow-up.

	Baseline Assessment	3-Month Follow-Up	Statistical	*p*-Value	Cohen’s d
**OTI-HSS ^a^**, *M(SD)*	3.90 (3.07)	1.28 (1.63)	t(141) = 9.825	0.000	0.86
**BDI ^b^**, *M(SD)*	6.03 (4.88)	5.05 (5.41)	t(141) = 2.22	0.027	0.20
**STAI-state ^c^**, *M(SD)*	17.32 (10.06)	17.68 (10.95)	t(141) = −0.410	0.682	0.03
**HRQOLDA ^d^**, *M(SD)*	78.49 (14.39)	91.79 (12.58)	t(141) = −11.625	0.000	0.92

OTI-HSS ^a^: Opiate Treatment Index-Health Symptom Scale; BDI ^b^ = Beck Depression Inventory; STAI-state ^c^ = State–Trait Anxiety Inventory; HRQOLDA ^d^ = The Health-Related Quality of Life for Drug Abusers Test.

**Table 3 ijerph-18-07442-t003:** Comparison between scores of the quality of life-related variables three months after admission to the therapeutic community and treatment outcomes (clinical impression) at discharge.

	Follow-Up Total Sample (*n* = 142)	Treatment Outcomes (Clinical Impression) at Discharge		Statistical	*p*-Value
		Non-Clinically Relevant Outcomes (*n* = 41)	Clinically Relevant Outcomes (*n* = 101)		
**OTI-HSS ^a^**, ***M(SD)***	**1.28 (1.63)**	**1.46 (2.13)**	1.21 (1.38)	t(140) = 0.845	0.399
**BDI ^b^**, *M(SD)*	5.05 (5.41)	6.56 (5.49)	4.44 (5.29)	t(140) = 2.145	0.034
**STAI-state ^c^**, *M(SD)*	17.68 (10.95)	21.12 (11.90)	16.29 (10.27)	t(140) = 2.425	0.017
**HRQOLDA ^d^**, *M(SD)*	91.79 (12.58)	87.90 (15.19)	93.37 (11.05)	t(140) = −2.384	0.041

OTI-HSS ^a^: Opiate Treatment Index-Health Symptom Scale; BDI ^b^ = Beck Depression Inventory; STAI-state ^c^ = State–Trait Anxiety Inventory; HRQOLDA ^d^ = The Health-Related Quality of Life for Drug Abusers Test.

**Table 4 ijerph-18-07442-t004:** Comparison between reliable changes in quality of life-related variables at the 3-month assessment and treatment outcomes (clinical impression) at discharge.

	Follow-Up Total Sample (*n* = 142)	Treatment Outcome Based on Clinical Impression at Discharge	
		Non-Clinically Relevant Outcomes (*n* = 41)	Clinically Relevant Outcomes (*n* = 101)
**OTI-HSS ^a^**, *n* (%)			
*Worsened ^b^*	11 (7.7)	5 (12.2)	6 (5.9)
*Unimproved ^c^*	18 (12.7)	6 (14.6)	12 (11.9)
*Improved ^d^*	40 (28.2)	9 (22.0)	31 (30.7)
*Recovered at cut-off point of 1SD ^e^*	45 (31.7)	10 (24.4)	35 (34.7)
*Recovered at cut-off point of 2SD ^f^*	28 (19.7)	11 (26.8)	17 (16.8)
**BDI ^g^**, *n* (%)			
*Worsened*	47 (33.1)	16 (39.0)	31 (30.7)
*Unimproved*	15 (10.6)	3 (7.3)	12 (11.9)
*Improved*	27 (19.0)	9 (22.0)	18 (17.8)
*Recovered at cut-off point of 1SD*	18 (12.7)	6 (14.6)	12 (11.9)
*Recovered at cut-off point of 2SD*	35 (24.6)	7 (17.1)	28 (27.7)
**STAI-state ^h^**, *n* (%)			
*Worsened*	66 (46.5)	23 (56.1)	43 (42.6)
*Unimproved*	7 (4.9)	1 (2.4)	6 (5.9)
*Improved*	28 (19.7)	6 (14.6)	22 (21.8)
*Recovered at cut-off point of 1SD*	22 (15.5)	7 (17.1)	15 (14.9)
*Recovered at cut-off point of 2SD*	19 (13.4)	4 (9.8)	15 (14.9)
**HRQOLDA ^i^**, *n* (%)			
*Worsened*	21 (14.8)	7 (17.1)	14 (13.9)
*Unimproved*	3 (2.1)	1 (2.4)	2 (2.0)
*Improved*	31 (21.8)	10 (23.4)	21 (20.8)
*Recovered at cut-off point of 1SD*	36 (25.4)	11 (26.8)	25 (24.8)
*Recovered at cut-off point of 2SD*	51 (35.9)	12 (29.3)	39 (38.6)

OTI-HSS ^a^: Opiate Treatment Index-Health Symptom Scale; Worsened ^b^: negative change in scale score; Unimproved ^c^: no change on the scale; Improved ^d^: positive change in the scale with a Reliable Change Index (RCI) <1 Standard Deviation (SD); Recovered at cut-off point of 1 SD ^e^: positive change in the scale with a Reliable Change Index (RCI) >1 SD; Recovered at cut-off point of 2SD ^f^: positive change in the scale with a Reliable Change Index (RCI) >2 SD; BDI ^g^ = Beck Depression Inventory; STAI-state ^h^ = State–Trait Anxiety Inventory; HRQOLDA ^i^ = The Health-Related Quality of Life for Drug Abusers Test.

## Data Availability

The data presented in this study are available on the request from the corresponding author.

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
