# Peer review of "In-Treatment Changes in Quality of Life-Related Variables in Therapeutic Communities for Cocaine Abusers: Are These Changes Associated with Clinical Outcomes?"

_ijerph, 2021, doi:10.3390/ijerph18147442_

Round 1

Reviewer 1 Report

I would like to thank the Authors for all changes and improvements they have done. I have no further comments.

This manuscript is a resubmission of an earlier submission. The following is a list of the peer review reports and author responses from that submission.

Round 1

Reviewer 1 Report

Therapeutic community (TC) has shown positive outcomes for patients with addictions. However, changes in quality of life (QoL) during the first 3 months of TC admission have not been evaluated. The objective of the study to determine the relationship between changes in QoL and treatment outcomes at discharge. The authors measured changes in QoL of 142 cocaine-use-disorder patients admitted to their 6 local TCs in 4 domains: addiction-related QoL, state anxiety, depression, and general health status, based on the clinical impression of their therapist at the TC for 3 months. The authors found that health-related QoL improved during the study period and state anxiety-like and depressive-life symptoms were decreased at discharged, suggesting that improvement in QoL-related variables was associated with better treatment outcome from TC.

While the study was well written, the statistical analysis methods selected used in the study is the main concern. For example, correlation, such as Pearson, or linear regression analysis, such as GLM, should be used instead of using t-test when relationship (i.e., association) is studied. It is not clear why the authors select this analysis while claiming to evaluate the “association” between QoL-related variables and treatment outcomes as the objective of the study. Linear regression is recommended to study “association”. Correlation is recommended to study relationship. The authors consistently stressed the Cohen’d throughout the manuscript. And Yet Cohen’s d is only to examine the effective sample size. With 142 subjects, Cohen’s d values are not relatively important but the correlation coefficient values or regression with p-values are the key interests of the majority of scientific readers. Further, the wording of the analysis was vague. For example, in Table 3.4, it states “Association analysis” in the legend. The authors did not specify and name the method of the analysis. The authors used only Student t-test throughout.

Other concerns include:

The Abstract states one objective in 1 sentence. The last paragraph of the Introduction states two objectives. In short, there should be only 1 objective, which is, to determine the relationship between changes in QoL and treatment outcomes at discharge. How and what to measure the variables to achieve the objective is not an objective.

Line 65: “… it is becoming…” to … it becomes…

Line 66: Gap between the citation and the word “Moreover”

Materials and Methods:

Please separate two subsections for 2.1. Design and setting is one. Participants is another. In former, please provide all sources of the data from the six TCs. And how these data were centralized. In the l. Similar to the Result 3.1, please explicit the characteristics of the participants, such as numbers of males and females, age ranges of them, numbers of participants included from the six TCs, race/ethnicity, and the likes.

The sample size should be 142, not 226. According to the scope and  objective of the study, the number of the subjects who were included and completed 3-month follow-up treatment is the sample size. Drop-outs do not count. Please revise the number in the manuscript and mention the causes of the 84 dropouts.

Statistical analysis: Please revise accordingly.

Line 203: Please correct the symbol between “and” and “for”, so as the whole manuscript.

Results:

In Table 1, please state the meanings of the superscript letters in the legend. Also for Lifetime SUD, please indicate whether the SUDs, from heroin to stimulants, were diagnosed with DSM-5. Nowadays in reports, there have been misinterpretations that drug use in history was categorized as SUDs and yet the individual might not be accessed by DSM-5 in lifetime. That is, subjects classified with SUDs must be diagnosed by DSM standards.

Author Response

Authors' reply to Reviewer 1

Comments to the Author

Therapeutic community (TC) has shown positive outcomes for patients with addictions. However, changes in quality of life (QoL) during the first 3 months of TC admission have not been evaluated. The objective of the study to determine the relationship between changes in QoL and treatment outcomes at discharge. The authors measured changes in QoL of 142 cocaine-use-disorder patients admitted to their 6 local TCs in 4 domains: addiction-related QoL, state anxiety, depression, and general health status, based on the clinical impression of their therapist at the TC for 3 months. The authors found that health-related QoL improved during the study period and state anxiety-like and depressive-life symptoms were decreased at discharged, suggesting that improvement in QoL-related variables was associated with better treatment outcome from TC.

We want to thank the reviewer for his/her constructive considerations regarding our paper that substantially improved our work and its quality. These comments and observations have helped us improve the precision in writing the results of our study. We will try to argue our answers. We have answered each of the queries in blue letters in this document. In addition, modifications to the manuscript are made in the track changes mode in MS Word.

While the study was well written, the statistical analysis methods selected used in the study is the main concern. For example, correlation, such as Pearson, or linear regression analysis, such as GLM, should be used instead of using t-test when relationship (i.e., association) is studied. It is not clear why the authors select this analysis while claiming to evaluate the “association” between QoL-related variables and treatment outcomes as the objective of the study. Linear regression is recommended to study “association”. Correlation is recommended to study relationship. The authors consistently stressed the Cohen’d throughout the manuscript. And Yet Cohen’s d is only to examine the effective sample size. With 142 subjects, Cohen’s d values are not relatively important but the correlation coefficient values or regression with p-values are the key interests of the majority of scientific readers. Further, the wording of the analysis was vague. For example, in Table 3.4, it states “Association analysis” in the legend. The authors did not specify and name the method of the analysis. The authors used only Student t-test throughout.

Regarding these comments, we distinguish two aspects:

  1. We acknowledge that we have not been precise enough in naming the analyzes we have performed. Our statistic is strictly a "comparative analysis".

Therefore, we have made some changes throughout the new manuscript to clarify this aspect.

  1. a)

Where it said (page 1, lines 16 and 17 - Summary): "The objective of this study was to evaluate the association between these changes and the clinical results at discharge"

In the new manuscript it says: "The objective of this study was to determine the relationship between these changes and treatment outcomes at discharge"

b)

Where it said (page 7, line 257); "Association analysis: ..."

In the new manuscript it says: "Comparison analysis: ..."

c)

Where it said (page 7. Line 259); "... association ..."

In the new manuscript it says. "... Comparison ..."

d)

Where it said (page 7. Line 265 - Table 3); "Association ..."

In the new manuscript it says: "Comparison ..."

e)

Where it said (page 7. Line 270); "Association analysis ..."

In the new manuscript it says: "Comparison analysis ..."

f)

Where it said (page 8. Line 289 - Table 4); "Association between ..."

In the new manuscript it says:  "Comparison between ..."

  1. With regard to the objection about the Cohen’d and its relation to the sample size, we understand the suggestion. However, we have performed this analysis and continue to maintain it because, together with the RCI of Jacobson and Truax, we believe that they are two clinically relevant change estimators to the research´s goal.

Other concerns include:

The Abstract states one objective in 1 sentence. The last paragraph of the Introduction states two objectives. In short, there should be only 1 objective, which is, to determine the relationship between changes in QoL and treatment outcomes at discharge. How and what to measure the variables to achieve the objective is not an objective.

We believe that it is a pertinent comment, in fact, this objection has been the result of debate between us and we have consulted a statistical mathematician. We finally admit that proposing a single goal is a clearer proposal and, above all, it is consistent with the objection of this same reviewer regarding the size of the sample. Therefore, we proceed to its modification. Although we admit that the way to measure the change of these variables is not properly an objective of the study, we are convinced that it is an important contribution of our manuscript.

Therefore, we have made some changes throughout the new manuscript to clarify this aspect.

Where it said (page 2. Line 70-75); " In this context, the present study had two main objectives. First, to evaluate reliable changes in four QoL-related variables (addiction-related QoL, state anxiety, depression, and general health status) in patients with cocaine use disorder after three months of treatment in a therapeutic community. Second, to determine the association between these changes and treatment outcomes at discharge based on clinical impression of their therapist in routine clinical practice.”

In the new manuscript it says:  "In this context, the main objective of this study was to determinate the relationship between changes in four QoL-related variables (addiction-related QoL, state anxiety, depression, and general health status) after three months of treatment and treatment outcomes at discharge in patients with cocaine use disorder in a therapeutic community..."

Line 65: “… it is becoming…” to … it becomes…

In the new manuscript we have corrected it

Line 66: Gap between the citation and the word “Moreover”

In the new manuscript we have corrected it

Materials and Methods:

Please separate two subsections for 2.1. Design and setting is one. Participants is another.

In the new manuscript we have corrected it

In former, please provide all sources of the data from the six TCs. And how these data were centralized. In the l. Similar to the Result 3.1, please explicit the characteristics of the participants, such as numbers of males and females, age ranges of them, numbers of participants included from the six TCs, race/ethnicity, and the likes.

Thanks for the comments. As described in “Procedures”, the external psychologist (E.M.) visited each TC and evaluated each patient. All the information was entered in the same database. As it is also described in the manuscript, the referral of patients from the outpatient centers to the TC is carried out by a single centralized office with a single criterion: optimization of waiting list times. For this reason, we do not expect relevant differences in the profiles of the patients in each TC. Therefore, this question does not seem relevant to the objectives of the study.

The sample size should be 142, not 226. According to the scope and objective of the study, the number of the subjects who were included and completed 3-month follow-up treatment is the sample size. Drop-outs do not count. Please revise the number in the manuscript and mention the causes of the 84 dropouts.

We agree with the reviewer and the new manuscript has been changed.

a)

Where it said (page 1, line 18-19-Abstract); " Prospective, longitudinal study of 226 cocaine-dependent patients treated at TCs in Spain. Of these patients, 142 (62.8%) reached the 3-month assessment and followed until discharge”

In the new manuscript it says: "Prospective longitudinal study of 142 cocaine-dependent patients treated at a therapeutics community. All of these patients reached the 3-month evaluation and were subsequently followed until discharge.”

  1. b)

Where it said (page 3, line 98-99): “The final sample included 227 patients. The sample size calculation was based on the data provided by Nunes et al. [55], assuming a power 1-b of 0.9 and a significance level of 0.05. The distribution of participants by treatment centre was as follows: Tarifa (22%), Almonte (21.6%), Los Palacios (21.5%), Cartaya (17%), La Línea (13.8%), and Mijas (4.1%). “

And where it said: “A total of 227 subjects were initially enrolled in the study. All eligible patients were invited to participate, all met the selection criteria, agreed to collaborate, and signed the informed consent. Of these 227 patients, one patient was excluded for failing to complete the self-administered QoL-related scales. Thus, the final sample included 226 participants (Table 1, column 1), most of whom were males (91.2%). The mean age was 34 years. Most participants were single (57.1%) with a secondary educational level (56.2%). The majority of participants were unemployed (67.3%) and 58.4% had a criminal record. Most patients (80.1%) had received prior treatment for substance use disorder and 43% had been previously treated for a mental disorder. In terms of the substance use disorder, most patients (67.3%) were speedball users (smoked/inhaled), while 37% were cocaine users (21.6% used powder cocaine and 11.1% freebased) without heroin use. The mean duration of cocaine use was 10 years, with a relatively severe dependence at admission. Other substance use disorders, in addition to cocaine dependence, included addiction to heroin (67.3%), cannabis (59.3%), and alcohol (58.8%). Nearly 70% of the patients had at least one co-occurring psychiatric disorder (dual diagnosis). The mean baseline scores ​on the QoL, state anxiety, depression and general health status scales indicated an average involvement for this patient population.”

In the new manuscript it says: “Considering the study objectives, the final sample included the 142 patients who reached the 3-month post-admission evaluation, out of a total of 226 subjects who met the study selection criteria. The sample size calculation was based on data provided by Nunes et al. [55], assuming a power (1-a) of 0.9 and a significance level of p ≤ 0.05. The distribution of participants by treatment centre was as follows: Tarifa (22%), Almonte (21.6%), Los Palacios (21.5%), Cartaya (17%), La Línea (13.8%), and Mijas (4.1%). All eligible patients were invited to participate and those who met the selection criteria, agreed to collaborate, and signed the informed consent were included. A total of 84 patients were excluded from the analysis due to failure to reach the minimum follow-up of 3-months; of these 84 patients, 61.9% (52), voluntarily withdrew from the TC and the other 38.1% were discharged early (32) for violating the rules of the TC. The sociodemographic and clinical characteristics of the final sample (n=142) are shown in Table 1, column 3. Most of the participants were males (92.3%). The mean age was 34.6 years. Most participants were single (64.1%) with a secondary educational level (58.5%). The majority of the participants were unemployed (68.3%) and 56.3% had a criminal record. Most patients (80.3%) had received prior treatment for substance use disorder and 40.8% had been previously treated for a mental disorder. In terms of the substance use disorder, most patients (66.4%) were speedball users (smoked/inhaled), while 33.6% were cocaine users (22.4% used powder cocaine and 11.2% freebased) without heroin use. The mean duration of cocaine use was 10 years, with a relatively severe dependence at admission. Other substance use disorders, in addition to cocaine dependence, included addiction to heroin (67.2%), cannabis (58.9%), and alcohol (58.3%). Two-thirds (66.2%) of the patients had at least one co-occurring psychiatric disorder (dual diagnosis). The mean baseline scores ​on the QoL, state anxiety, depression and general health status scales were consistent with the average scores typically observed in this patient population”.

Statistical analysis: Please revise accordingly.

According to the above, we believe that it is not appropriate to make these changes.

Line 203: Please correct the symbol between “and” and “for”, so as the whole manuscript.

In the new manuscript we have corrected it

Results:

In Table 1, please state the meanings of the superscript letters in the legend. Also for Lifetime SUD, please indicate whether the SUDs, from heroin to stimulants, were diagnosed with DSM-5. Nowadays in reports, there have been misinterpretations that drug use in history was categorized as SUDs and yet the individual might not be accessed by DSM-5 in lifetime. That is, subjects classified with SUDs must be diagnosed by DSM standards.

In the new manuscript we have corrected it

Reviewer 2 Report

The article is of interest and I hope the authors can solve some shortcomings: -It is advisable to avoid the use of acronyms in the abstract. -It is necessary that the introduction present greater coherence between the ideas because they sometimes appear without joining. It is recommended to expand the number of citations because sometimes the statements are free of them. - A further justification of the objectives of the study is needed. - More data is needed on the sociodemographic characteristics of the participants. Indicate the date on which the data were collected. - Please indicate information about the type of selection used. - The procedure has unanswered gaps. It is not clear when to administer the tests (indicate the date they were performed). - How likely is a subject to be part of the study? - Data on the characteristics of the sample are not part of the results. Please indicate that information in the relevant place. - The results are not clear in the section. It is recommended to present them in such a way that they respond to the objectives of the study. -The discussion is not rich and the main results of the study are not discussed. Please add more limitations. They are necessary because there are many methodological difficulties.

Author Response

Authors' reply to Reviewer 2

Comments to the Author

The article is of interest and I hope the authors can solve some shortcomings:

We want to thank the reviewer for his/her constructive considerations regarding our paper that substantially improved our work and its quality. These comments and observations have helped us improve the precision in writing the results of our study. We will try to argue our answers. We have answered each of the queries in blue letters in this document. In addition, modifications to the manuscript are made in the track changes mode in MS Word.

It is advisable to avoid the use of acronyms in the abstract.

We are agreeing with the review´s suggestion and in the new manuscript we change it.

It is necessary that the introduction present greater coherence between the ideas because they sometimes appear without joining.

We agree on the importance of having a good coherent structure of ideas in the introduction. Thus, we have made special interest in getting an introduction that synthesizes the state of the art and justifies the pertinence and relevance of our study. The introduction first, begins with a heading about epidemiological aspects. Second, it summarizes the evidence on the overall effectiveness of TC in the treatment of addictions. Third, it identifies the factors that determine or modulate this effectiveness, highlighting the relevant role played by the length of stay in the TC. Finally, the absence of studies that analyze the role of variables related to quality of life as variables associated with the response to treatment at discharge is noted. Therefore, we think that the Introduction follows a coherent line of argument.

It is recommended to expand the number of citations because sometimes the statements are free of them.

In the Introduction we have supported each idea with a citation. Some sentences do not end in a reference because two sentences in a row are supported by the same citation.

A further justification of the objectives of the study is needed.

Thanks for the suggestion, as we have already indicated above, the objectives of the study are adequately justified based on the verification in the literature of the absence of studies that consider the predictive role of the changes that occurred during the first 3 months of admission to CT and the results of treatment at discharge.

More data is needed on the sociodemographic characteristics of the participants.

We have measured those sociodemographic data that help to characterize the sample and we have included all the sociodemographic data that we have measured.

Indicate the date on which the data were collected.

The data was collected between 2008 and 2010.

Please indicate information about the type of selection used.

According with the review´s comment we have specified more this sentence:

Where it said : “A consecutive sampling techique was used to recruit participants for this study”

In the new manuscript it says: “A consecutive sampling technique was used to recruit participants for this study, including all patients who met the selection criteria”.

The procedure has unanswered gaps. It is not clear when to administer the tests (indicate the date they were performed).

This information appears in the original manuscript, specifically on page 4, line 189.

How likely is a subject to be part of the study?

The probability that a subject will participate in the study is determined by the sampling procedure (see page 2, line 93-94). The type of recruitment is detailed in the Procedures section, page 4.

Data on the characteristics of the sample are not part of the results. Please indicate that information in the relevant place.

We agree with this observation and have made the corresponding changes. All the information corresponding to “Sample characteristics” (page 5 in the original manuscript), now goes to the end of the new section “2.2. Participants. "

The results are not clear in the section. It is recommended to present them in such a way that they respond to the objectives of the study.

We agree with this observation. Given that reviewer 1 made a similar observation about the results, we have modified the text to address both comments. The results section has been renumbered from 3.1 to 3.5.

The discussion is not rich and the main results of the study are not discussed. Please add more limitations. They are necessary because there are many methodological difficulties.

We agree with to the review´s comment, we recognize that our study has several limitations. Therefore, we add the following sentence to the limitations section (within Discussion):

“Third, it is important to keep in mind that we used self-administered scales (e.g., BDI and STAI) in this study to evaluate depression and anxiety. Therefore, we cannot rule out a subjectivity bias. To reduce the risk of this type of bias, future studies should include objective assessment instruments such as the Hamilton Depression Rating Scale (HDRS) and/or the Hamilton Anxiety Rating Scale (HDRS), which are clinician-administered scales.”

Reviewer 3 Report

In general the study is very interesting and it covers an important topic of the quality of life of cocain abusers treated in TC`s. However some revision of the manuscript is necessary:

  1. I need some clarification about the criteria of the admission to the TC - Did the patients udergo any detoxification prior to the admission or after the admission?
  1. In Materials and Methods it is mentioned that in all participating centers a pharmacological treatment is offered (methadone or buprenorphine). Did the participants recruited to the study use this type of medication? And if yes- for how long time?
  2. In the Results it was reported that "Nearly 70% of the patients had at least one co-occurring psychiatric disorder (dual diagnosis)". What kind of comorbid psychiatric disorders the patient suffered from? Did they receive any medication (antidepressants, antipsychotics) because of that?
  3. In the discussion of limitations, in my opinion it should be added that using such tools like BDI and STAI may bring subjective results of the assessment of the patient`s clinical state, because they are self-administered. For an objective evaluation scales like HDRS would be better.

Author Response

Authors' reply to Reviewer 3

Comments to the Author

In general, the study is very interesting and it covers an important topic of the quality of life of cocaine abusers treated in TC`s. However, some revision of the manuscript is necessary

We want to thank the reviewer for his/her constructive considerations regarding our paper that substantially improved our work and its quality. These comments and observations have helped us improve the precision in writing the results of our study. We will try to argue our answers. We have answered each of the queries in blue letters in this document. In addition, modifications to the manuscript are made in the track changes mode in MS Word.

I need some clarification about the criteria of the admission to the TC.

As we described in the original manuscript, the public TCs in Andalucia for the treatment of addictions are part of a network of healthcare services, in which outpatient treatment centers are integrated. The decision for referral to TC scans is made by clinicians at outpatient treatment centers. Thus, TCs do not have own admission criteria. A centralized management public office receives all requests for referral to TC and prioritizes them based on the availability of free places

Did the patients undergo any detoxification prior to the admission or after the admission?

No, the public TCs in Andalucía have medical and nursing staff to perform detoxification treatments within the TC. The goal is for the patient to enter TC rehabilitation activities as soon as possible.

In Materials and Methods it is mentioned that in all participating centers a pharmacological treatment is offered (methadone or buprenorphine). Did the participants recruited to the study use this type of medication? And if yes- for how long time?

We acknowledge that this phrase has led to a misunderstanding. What we mean is that drug treatment for his addiction is also offered to the patient. Patients with opioid addiction with / without other associated addiction can receive treatment with methadone or buprenorphine within TC. To avoid this misunderstanding, we made the following modification to the manuscript:

Where it said:  All of these centres offer the same treatment programme based on a multidisciplinary approach, consisting of cognitive-behavioral therapy, psychoeducation, occupational therapy, and pharmacological treatment (methadone or buprenorphine)”

In the new manuscript it says: “All of these centres offer the same treatment programme based on a multidisciplinary approach, consisting of cognitive-behavioral therapy, psychoeducation, occupational therapy, and pharmacological treatment”.

In the Results it was reported that "Nearly 70% of the patients had at least one co-occurring psychiatric disorder (dual diagnosis)". What kind of comorbid psychiatric disorders the patient suffered from? Did they receive any medication (antidepressants, antipsychotics) because of that?

The distribution of frequencies of mental disorders throughout life was as follows: mood disorders 31.3%, anxiety disorders 23.8%, psychotic disorders 18.9%, eating disorders 1.8%, antisocial personality disorder 22.5%, personality limit 14.1% and attention deficit hyperactivity disorder 14.5%. Some of these patients were already diagnosed before admission to TC and were undergoing treatment for their mental disorder. Other patients were diagnosed on TC during the evaluation period and treatment for their mental disorder was proposed.

In the discussion of limitations, in my opinion it should be added that using such tools like BDI and STAI may bring subjective results of the assessment of the patient`s clinical state, because they are self-administered. For an objective evaluation scales like HDRS would be better."

We agree with this objection. Therefore, we add the following sentence to the limitations section (within Discussion):

Third, it is important to keep in mind that we used self-administered scales (e.g., BDI and STAI) in this study to evaluate depression and anxiety. Therefore, we cannot rule out a subjectivity bias. To reduce the risk of this type of bias, future studies should include objective assessment instruments such as the Hamilton Depression Rating Scale (HDRS) and/or the Hamilton Anxiety Rating Scale (HDRS), which are clinician-administered scales.”
